# The Evolution and Outcomes of a Collaborative Testbed for Predicting Coastal Threats

**Charles Reid Nichols [1] and Lynn Donelson Wright [2,*]**

[1]   Marine Information Resources Corporation, Ellicott City, MD 21042, USA; rnichols@mirc-us.com
[2]   Southeastern Universities Research Association, Washington, DC 20005, USA
*   Correspondence: Wright@sura.org

**Abstract:** Beginning in 2003, the Southeastern Universities Research Association (SURA) enabled an open-access network of distributed sensors and linked computer models through the SURA Coastal Ocean Observing and Predicting (SCOOP) program. The goal was to support collaborations among universities, government, and industry to advance integrated observation and modeling systems. SCOOP improved the path to operational real-time data-guided predictions and forecasts of coastal ocean processes. This was critical to the maritime infrastructure of the U.S. and to the well-being of coastal communities. SCOOP integrated and expanded observations from the Gulf of Mexico, the South Atlantic Bight, the Middle Atlantic Bight, and the Chesapeake Bay. From these successes, a Coastal and Ocean Modeling Testbed (COMT) evolved with National Oceanic and Atmospheric Administration (NOAA) funding via the Integrated Ocean Observing System (IOOS) to facilitate the transition of key models from research to operations. Since 2010, COMT has been a conduit between the research community and the federal government for sharing and improving models and software tools. SCOOP and COMT have been based on strong partnerships among universities and U.S. agencies that have missions in ocean and coastal environmental prediction. During SURA's COMT project, which ended September 2018, significant progress was made in evaluating the performance of models that are progressively becoming operational. COMT successes are ongoing.

**Keywords:** numerical models; model comparisons; metadata; model coupling; ocean observations; scientific collaboration; model testbed; hurricanes; storm surge; hypoxia

## 1. Introduction

Environmental modeling complements long-term ocean observation programs; together, these are helping scientists to better understand climate and marine ecosystems, as well as human impacts and vulnerabilities [1]. More effective prediction, communication, mitigation, and response to coastal processes on multiple time scales are essential to the welfare of coastal communities and to the sustainability of coastal ecosystems and human infrastructure [2–4]. Validation of model predictions requires continuous long-term data collection and continuing refinement and testing of model codes to ensure that performance meets or exceeds benchmarks [5]. These are some of the reasons that coastal ocean observatories have emerged as integrated, operational suites of instruments, permanently deployed to measure meteorological, oceanographic, and geophysical phenomena. Development of the National Oceanic and Atmospheric Administration (NOAA) U.S. Integrated Ocean Observing System (IOOS®) and linked Regional Associations has improved the capability of the scientific and operational oceanography community to monitor and forecast environmental changes and hazardous events.

By way of the Southeastern Universities Research Association (SURA) Coastal Ocean Observing and Prediction Program (SCOOP) and the subsequent Coastal Ocean Model Testbed (COMT) project, numerous collaborators have advanced ocean modeling programs supported by observing technologies.

The overarching scientific goals that that have guided these collaborative coastal and environmental science pursuits include the following:

Goal 1—Enable discovery of diverse and transdisciplinary coastal phenomena;

Goal 2—Couple observation and modeling of processes across science domains;

Goal 3—Enable high-resolution studies of multi-scale coastal phenomena;

Goal 4—Advance information and predicting services for basic and applied scientific research and innovative education and outreach.

This paper outlines the most prominent results of coastal modeling testbed efforts over the past 17 years that have advanced our knowledge of coastal ocean phenomena though observations and improved application of models. This experience now guides ongoing progress towards a fully operational system of coastal ocean models for high-resolution forecasts that include the influence of climatic and ocean processes downscaled to the coastal ocean.

## 2. SCOOP: A Launch Pad for Testbed Development

Observation and modeling programs such as SCOOP have improved ocean prediction, adaptive sampling, and observational system design [6,7]. The SCOOP open-access network of sensors and linked, web-accessible computer models guided coastal stewardship, enabled planning for extreme events, facilitated safe and efficient maritime operations, and supported military littoral security [8,9]. SCOOP provided access to simultaneous measurements of winds, waves, currents, water density, nutrients, water quality, and biological indices under all conditions. SCOOP facilitated a community modeling approach that was distinct from traditional approaches by a single research organization [10]. The result was a community of research institutions and operational agencies that turned observations and predictions into environmental intelligence. SCOOP also demonstrated that the relationship between basic research and applications can and must be bidirectional. Research by operational projects must assess and demonstrate scientific model skill to support operational needs, while simultaneously conducting critical assessments to improve current operational models such as the sea, lake, and overland surges from hurricanes (SLOSH) model [11].

The adoption of rigorous Open Geospatial Consortium (OGC) standards was crucial to SCOOP and subsequent testbed efforts to ensure data quality and accessibility through open interfaces [12–16]. The OGC (http://www.OpenGeospatial.org) is composed of more than 500 members from government (including NOAA), commercial organizations, NGOs, and academic and research organizations. Standards developed by the OGC were instrumental in achieving the aforementioned scientific goals. The OGC Web Mapping Service (WMS) standard was used to exchange satellite images, hurricane tracks, and model output. SCOOP participants also used the OGC Sensor Observation Service (SOS) standard to share real-time sensor information from different regional observatories. SCOOP led an interoperability experiment within the OGC Innovation Program that helped to advance a new revision of the SOS standard benefiting ocean observing systems around the world [17]. SCOOP improved the discoverability, accessibility, and usability of marine sensors, transducers, and sensor data repositories.

### 2.1. SCOOP Products and Accomplishments

The SCOOP program began in 2003 and included participants from academia, the private sector, and government agencies. The four pillars of SCOOP were (1) basic and applied research, (2) technology transition, (3) archive of data and enablement of discoverable information, and (4) dissemination of data using open standards for encoding along with interfaces and software tools. The major SCOOP accomplishments summarized below advanced ocean observations in the United States and Canada [18–21].

### 2.1.1. Coastal Modeling

The SCOOP architecture consisted of several modules that enabled distributed modeling of the Atlantic and Gulf of Mexico coasts [22]. Coastal wave, surge, and hydrodynamic model runs utilized

WaveWatch III® [23], Eulerian–Lagrangian CIRCulation (ELCIRC) model [24], Simulating WAves Nearshore (SWAN) [25]; WAve Model (WAM) [26]; Curvilinear Hydrodynamics in Three Dimensions (CH3D) [27], and ADvanced CIRCulation (ADCIRC) [28]. Model runs were performed on grid-enabled high-performance computing resources. The SURAgrid network provided a uniform portal to address diverse applications. For example, a multi-member wave model ensemble was run on grid-enabled computers at Louisiana State University (LSU) and the University of Florida. Storm surge simulations were run during Hurricane KATRINA and the results, along with observed waves, were made available in real time on the SCOOP "OpenIOOS" website.

### 2.1.2. Data Stewardship

The SCOOP program emphasized web-based mapping standards and geographic information systems (GIS) applications that displayed high-resolution base layers. The OGC Interoperability Experiment completed in 2007 provided a wealth of guidelines and best practices examples that allow widespread community participation in interoperable data sharing of various in-situ observation data for the U.S. IOOS. SCOOP implemented open standards that formed the foundation for the discovery, use, and integration of scientific data sets. Visualization infrastructure included the automatic transfer of hurricane-modeled forecasts to GIS formats [29,30]. The combined tasks provided data discovery and access capabilities for the whole SCOOP community.

### 2.1.3. Computer Infrastructure

SURA grid allowed participating organizations to share high performance computing (HPC) resources available on the "grid" to provide an opportunity and mechanism to share excess computational capabilities on one campus with other participants for research needs [31]. Furthermore, an OpenIOOS website primarily served as a community-building tool and a technology demonstration for the ocean observing community. The infrastructure capability delivered data using the OGC Web Feature Service (WFS), which included quality assurance/quality control procedures that ensured the highest quality data and reliability. Computer infrastructure was key to collaboration and, in addition to software, included job scheduling and management, storage, and hardware for the SCOOP grid. Middleware requirements were identified and developed to support a number of HPC needs such as dynamic scheduling, allocations, and resource provisioning [12].

### 2.1.4. Community Engagement

A SCOOP Education Virtual Appliance developed by the University of Florida enhanced the ability of scientists and non-scientists to understand the primary factors influencing coastal and estuarine processes. SCOOP partnered with the Gulf of Mexico Coastal Ocean Observing System and this interaction was critical in transferring research results to benefit those running GNOME (General NOAA Operational Modeling Environment), a modeling tool used by NOAA's Office of Response and Restoration [11].

During SCOOP, a grid-enabled archive was designed and deployed at LSU. Concomitant data management services were critical to supporting the researchers. The archive ingested files arriving over different protocols, and a client tool was developed using a grid application toolkit, which could retrieve files using either File Transfer Protocol for grid computing (GRIDFTP) or HTTP. Queries to the archive were made based on metadata [32]. Ensemble modeling, querying, and downloading files from the archive and file transfers were among the capabilities facilitated by the SCOOP grid. The Gulf of Mexico grid was tested in real time during Hurricane Katrina [12].

From 2003 through 2009, SCOOP Program participants produced over 40 publications, including student theses, book chapters, journal articles, and conference proceedings. Modeling findings included the need for more data, new sensors, and expanded observational programs, especially in the nearshore. Some examples include the need for more wave observations to improve wave forecasts [33]. Improved process representation, better model coupling, incorporation of data assimilation techniques,

and testing of real-time models were identified as future requirements at the same time that the emerging U.S. IOOS was beginning to provide the foundation needed to establish a testbed to compare and improve models.

## 2.2. Collaboration during SCOOP

SURA established a viable structure to enable partners to comprise SCOOP, a big science project. A leader from SURA was selected to facilitate the sharing of data and the integration of multi-disciplinary capabilities among the partners listed in Table 1. SURA utilized its resources to create an environment for successful interaction among disparate scientists. The groundwork was laid for collaboration by facilitating scientific interaction on numerous levels, ranging from project meetings to annual all-hands meetings. SURA leaders ensured that resources could be focused on novel results that were grounded in fundamental marine science and technology. Team science culminated in improved use of historical information, observations, and ultimately improved forecasts. These activities contributed to the eventual implementation of a viable coastal ocean prediction system.

**Table 1.** Southeastern Coastal Ocean Observing and Predicting Program (SCOOP) collaborators and partners.

| University | Government | Industry/NGO |
|---|---|---|
| Louisiana State University; Texas A & M University; University of Florida; University of South Florida; University of Alabama in Huntsville; University of North Carolina at Chapel Hill, and Virginia Institute of Marine Science of the College of William & Mary | Bedford Institute of Oceanography (CA); Office of Naval Research, and National Oceanic and Atmospheric Administration (NOAA) | Gulf of Maine Ocean Observing System; Gulf of Mexico Coastal Ocean Observing System, and the Southeastern Universities Research Association |
| Funding support for SCOOP was provided by the Office of Naval Research, Award N00014-04-1-0721, and NOAA National Ocean Service, Award NA04NOS4730254. | | |

## 3. Coastal Ocean Model Testbed (COMT): Concept and Goals

Like its precursor, SCOOP, COMT has provided direct access to data and models for developing and testing model improvements and communicating results to forecasters and users. The long-range vision from the outset of this program has been to increase the accuracy, reliability, and scope of operational coastal and ocean forecasting products and to be responsive to the needs and concerns of operational end users. Through this program, SURA helped to facilitate technology transition (e.g., numerical models) to operational organizations such as the Navy and NOAA. These operational organizations, while collaborating with academic researchers, also benefitted from the collaborative environment and ongoing university research to improve operational models, tools, and products.

The SCOOP program demonstrated the importance of using numerical models to improve predictions of the impacts of events such as Hurricane Katrina. Table 2 provides general information related to models developed by community modelers throughout the United States and Canada. Assessments of the ability of various models to predict ocean properties and circulation was considered essential for operational organizations such as NOAA. The intent of COMT was to build on SCOOP and develop a procedure to utilize new generations of models to aid our understanding of extreme events such as inundation and chronic environmental conditions such as hypoxia.

Consistent with NOAA's goals, the early COMT program was intended to enable participation by the coastal ocean modeling community in evaluating and improving models that might ultimately become operational. The COMT infrastructure and programs allowed the participants to share numerical model code, results, observations, software tools, and common evaluation methodologies. The aim was to aid the transition of promising models to operational use in addressing issues such as inundation and hypoxia. Collaboratively written project materials from semiannual progress reports and annual meetings are accessible online at https://comt.ioos.us/. Figure 1 illustrates the pathways by which COMT research activities were transitioned to operations [34].

**Table 2.** Circulation models applied in the Coastal Ocean Model Testbed (COMT) program used various numerical formulations including finite differences, finite elements, finite volumes; implicit and explicit time stepping and various vertical coordinate treatments (e.g., vertically integrated, z-level, terrain following, isopycnic, and hybrid). Wave models were based on a random-phase, spectral description of the ocean surface.

| Acronym | Model Name | Description |
|---|---|---|
| ADCIRC | ADvanced CIRCulation | A finite element unstructured grid model. |
| CH3D | Curvilinear-Grid Hydrodynamics 3D | A finite-difference three-dimensional model applicable to bays, estuaries, lakes, and rivers. |
| EFDC | Environmental Fluid Dynamics Code | A linked three-dimensional, finite difference hydrodynamic and water quality model. |
| FVCOM | Finite Volume Community Ocean Model | Unstructured-grid, finite-volume, 3-D coastal ocean circulation model |
| HYCOM | Hybrid Coordinate Ocean Model | A general circulation model. |
| NCOM | Navy Coastal Ocean Model | A high-resolution model adapted from the Princeton Ocean Model and the Sigma/Z-Level Model. |
| ROMS | Regional Ocean Modeling System | Terrain-following ocean model. |
| SELFE | Semi-implicit Eulerian–Lagrangian Finite Element | A finite-element model for cross-scale ocean modeling. |
| SWAN | Simulating WAves Nearshore | A spectral wave model developed at the Delft University of Technology in the Netherlands. |
| WWM | Wind Wave Model | A spectral wave model similar to SWAN but unstructured grid. |
| WW3 | WaveWatch III® | A third-generation wave model developed at NOAA/National Center for Environmental Prediction (NCEP). |

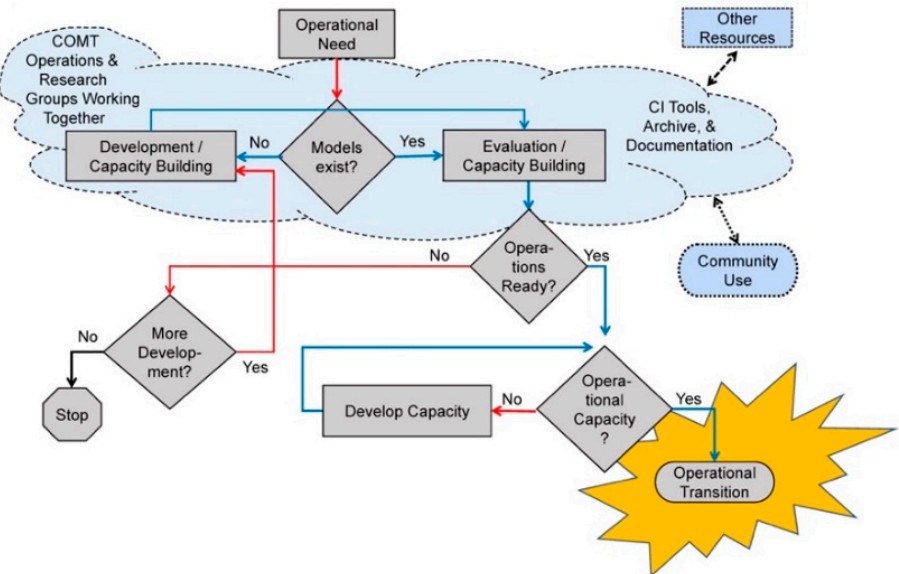

**Figure 1.** Flow diagram illustrating the connections among the components and outcomes of COMT. Redrawn based on [34] with permission from Journal of Geophysical Research, 2020. The original intent of COMT was to transfer new or improved scientific knowledge or technologies produced by research to operations. The development of transition pathways as COMT progressed were described by [34].

### 3.1. COMT: Phase 1

The initial phase of the COMT program ran from 2010 to 2013. Projects focused on modeling phenomena affecting the U.S. Atlantic and Gulf of Mexico coasts. The emphasis was on inundation, estuarine hypoxia in the Chesapeake Bay, shelf hypoxia in the Northern Gulf of Mexico,

and development of improved tools for model assessment. SURA provided the COMT program with collaboration leadership to facilitate sharing of HPC resources and knowledge transfer, especially implementation of advances made by the cyberinfrastructure team. During this initial phase, COMT provided guidance on the accuracy, robustness, execution speed, and implementation requirements (e.g., resolution, parameterization, computer capacity) of models such as ADCIRC, FVCOM, SELFE, SWAN, and WaveWatch III. Models were evaluated for eventual operational use by "hind-cast" comparison with data from the National Data Buoy Center; NOAA tide gauges; USGS permanent and temporary gauges; the Gulf of Mexico Coastal and Ocean Observing System, and other complementary sources from Hurricane Bob (1991) in the Gulf of Maine and Hurricanes Rita (2005) and Ike (2008) in the Gulf of Mexico.

Identical grids and forcing were used in comparisons of the coupled unstructured surge-wave models, ADCIRC+SWAN, FVCOM+SWAN, and SELFE+WWM, in runs carried out for Hurricanes Rita and Ike. The Galveston Basin grid, along with the Gulf of Mexico grid and a local grid for Galveston and Sabine Pass, were used for runs of SLOSH, which failed to capture the Hurricane Ike forerunner and underestimated storm surge height [35]. The three unstructured models performed much better at reproducing observed processes, yielded similar results, and showed variances comparable to observations. SLOSH deviated more from observations than did the unstructured grid models in all cases. As expected, the runtime for SLOSH was much faster than was the case for the other models. All of the models reproduced important oceanographic processes, with minimal water level fluctuation bias and comparable variances versus observations. SLOSH performed better on the Sabine Pass Basin grid for Hurricane Rita owing to differences in forerunner surge, which is generally caused by shore-parallel winds and resulting currents [36]. On the Extra-Tropical Storm Surge (ETSS) grid, SLOSH showed minimal bias for either storm, although its scatter was quite large due to the nearly 5 km resolution in coastal areas. As indicated in progress reports (see https://comt.ioos.us/), SLOSH differences from observed data were greater than those from the three unstructured grid models. The largest deviation in model performance was observed in execution speed and scalability. The implicit time stepping scheme of SELFE coupled to WWM performed well at small numbers of cores but scaled poorly at larger numbers of cores. Coupled ADCIRC and SWAN computing infrastructure was described for comparisons of Hurricanes Katrina and Rita [37].

ADCIRC coupled to SWAN during COMT resulted in improved scaling and absolute performance when more than 128 cores were used per run. SLOSH, which runs on a single core, is not configured to utilize modern parallel computing architecture. The COMT project investigators reported that runtimes for ADCIRC on a single core were more than 10 times longer than for SLOSH on the ETSS grid, even after the SLOSH runtimes were normalized for the number of grid nodes (https://comt.ioos.us/). SLOSH, as described by [38], is used operationally to combine large ensembles of model runs for probabilistic flood forecasting. However, the accuracy of SLOSH-based probabilistic forecasts should be assessed by comparing select, individual SLOSH runs with similar runs using higher resolution unstructured grid models, as discussed above. A description of the program from its inception and a compilation of scientific results is presented in a series of 16 scientific papers in a special issue of the Journal of Geophysical Research [34].

*3.2. COMT: Phase 2*

The most recent COMT projects began in 2013 and included participants from academia, the private sector, and government agencies (see Table 3). Phase 2 COMT projects further advanced the operational use of models for the prediction of extreme events and chronic conditions [39]. Of particular importance was modeling of low or depleted oxygen (hypoxia) and the storm surge inundation of coastal areas adjacent to steep sloped bathymetry. Table 3 lists the Phase 2 participants.

**Table 3.** Coastal and Ocean Modeling Testbed collaborators and partners. Collaborators were funded to work on projects. Most of the partners from government or the U.S. IOOS Regional Associations joined project teams in pursuit of the shared goal to transition new modeling capabilities to operations.

| University | Government | Industry/NGO |
| --- | --- | --- |
| Dalhousie University; Louisiana State University; Oregon State University; Texas A & M University; University of California San Diego; University of California Santa Cruz; University of Florida; University of Maine; University of Maryland; University of Massachusetts-Dartmouth; University of North Carolina; University of Notre Dame; University of Puerto Rico; University of South Florida; University of Washington; Virginia Institute of Marine Science of the College of William & Mary; Woods Hole Oceanographic Institution | NOAA Coast Survey Development Laboratory, Environmental Modeling Center, National Hurricane Center, Naval Research Laboratory-Oceanography Division; U.S. Army Corps of Engineers—Coastal & Hydraulics Laboratory; U.S. Environmental Protection Agency-Gulf Ecology Division; U.S. Integrated Ocean Observing System | Delta Modeling Associates; Remote Sensing Solutions; RPS-Applied Science Associates; Southeastern Universities Research Association |

**Partners:** Southern California, Central and Northern California, Pacific Northwest, Mid-Atlantic, Gulf of Mexico, and Caribbean IOOS Regional Associations; IOOS Association; Chesapeake Bay Interpretive Buoy System; Environmental Protection Agency; National Environmental Satellite, Data, and Information Service; National Ocean Service; National Weather Service, and United States Geological Survey.

Funding support for COMT was provided by NOAA's National Ocean Service under Cooperative Agreement NA04NOS4730254, NA10NOS0120063, NA11NOS0120141, and NA13NOS0120139.

The five projects of COMT Phase 2 are summarized as follows.

(1)    Estuarine Hypoxia Modeling in Chesapeake Bay

The Chesapeake Bay receives and transforms nutrients from tributaries such as the Susquehanna, Potomac, Rappahannock, York, and James rivers before they are exported to the adjacent continental shelf. Although the water quality of the Chesapeake Bay has been estimated from in situ observations [40,41], uncertainties associated with inter-annually varying hydrological conditions persist. For these reasons, the estuarine hypoxia project focused on quantifying the temporal and spatial variability of low and harmful dissolved oxygen (DO) concentrations using a variety of models that were adapted from ROMS, EFDC, and CH3D for the Chesapeake Bay. The project utilized scenario-based modeling of hypoxia in the Chesapeake Bay and compared the skill of multiple Chesapeake Bay models to predict salinity, temperature, stratification, and DO concentration. Data for inter-comparisons with model output were derived from the Chesapeake Bay Interpretive Buoy System (CBIBS) and satellite imagery. After completing a multiple-model inter-comparison of nine different models of the bay, the project concluded that the use of multiple models should support decision-making by managers. Most models were found to have difficulty resolving key drivers (winds, tides, solar radiation) of DO variability, but all the models exhibited some level of skill to estimate DO variability. Simple oxygen models and complex biogeochemical models all reproduced observed DO variability well, but the ensemble reproduced the observations better than any individual model. Successes have helped to determine the role of estuarine processes and climate change in enhancing or mitigating DO concentrations [42–45]. As indicated in Figure 2, forecasts of DO in the Chesapeake Bay are made available to the public through the Mid-Atlantic Regional Association Coastal Ocean Observing System (MARACOOS) Oceans Map (http://oceansmap.maracoos.org/) and through the Virginia Institute of Marine Science (VIMS).

The estuarine hypoxia research included collaborators from VIMS, University of Maryland Center for Environmental Science, Woods Hole Oceanographic Institution, and Delta Modeling Associates. Technology transfer from this project is being planned through MARACOOS and ultimately to enhance NOAA's Chesapeake Bay Operational Forecast System. Research advances were described during a stakeholder workshop that was held at VIMS on 26 April 2016. Future improvements in hypoxia simulations will depend on advances in the ability to model the depth of the mixed layer.

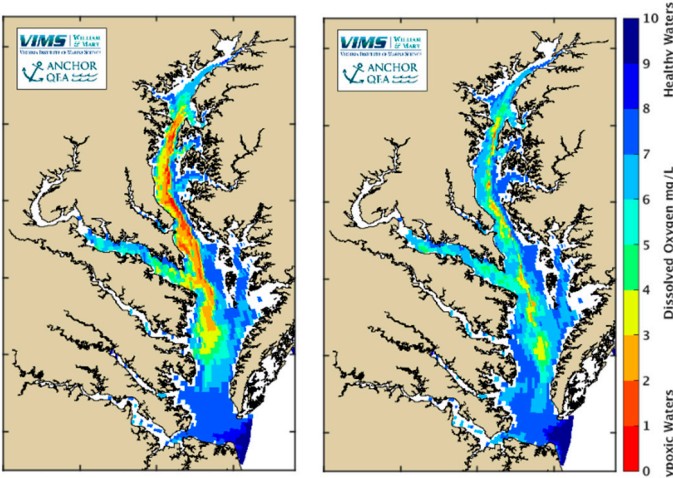

**Figure 2.** Bottom oxygen nowcast (left panel) from 24 July 2018 and bottom oxygen forecast for 26 July 2018 during a period of high precipitation that caused some minor flooding along the upper and middle Chesapeake Bay regions. Hypoxia forecasts provide a useful result that supports users that are interested in the extent, duration, and severity of low-oxygen dead zones in the Chesapeake Bay. For more detail, see http://www.vims.edu/research/topics/dead_zones/forecasts/cbay/index.php. Public Domain Figure.

(2)    U.S. West Coast Physics and Ecosystems Modeling

This project supported the development of an operational nowcast and forecast hydrodynamic modeling system for the west coast of the United States. This required integration of observing system data streams, hydrodynamic model predictions, product dissemination, and continuous quality-control monitoring. In support of operational partners from NOAA, this project focused on methods for data assimilation in order to improve forecast accuracy and inter-comparison of the biogeochemical models that could provide useful prediction of productivity, oxygen concentrations, and acidity in coastal waters and involved the creation of a high-resolution regional model [46]. The study helped to explain anomalous warming along the Northern California Coast [47] and facilitated inter-comparison of quasi-operational regional and coastal ocean modeling systems supported by the three IOOS Regional Associations along the West Coast, i.e., Southern California Coastal Ocean Observing System, Central and Northern California Ocean Observing System, and the Northwest Association of Networked Ocean Observing Systems. The data assimilation relied heavily on NOAA satellite observations of sea level and sea surface temperature, land-based high-frequency radars measuring surface currents over the shelf, and in-situ observations collected by IOOS Regional Associations [48–50]. Model forecast results are available on selected Regional Association websites and LiveOcean (see https://faculty.washington.edu/pmacc/LO/LiveOcean.html). The best methods and practices from the COMT were transitioned to the NOAA West Coast Operational Forecast System (WCOFS) in testing at the NOAA National Oceanographic Service.

(3)    Surge and Wave Modeling for Puerto Rico/U.S. Virgin Islands

Research along the steep-sloped Caribbean coast in COMT expanded on the coastal inundation modeling along shallow shelves completed during the first phase of COMT. Despite being in the pathway of Atlantic hurricanes and having significant coastal populations and infrastructure, little modeling has been done on inundation in Puerto Rico and the U.S. Virgin Islands. The finite element ADCIRC, third generation WaveWatch III, spectral SWAN, and operational SLOSH models were selected for this project. These models were evaluated for Hurricane George (1998), Hurricane Isaac (2012), and Superstorm Sandy (2012). Project results demonstrated the critical need for a coupled surge and wave model and led to the development of a wave model compatible with the SLOSH surge model.

A surge atlas was created for the affected islands and used by emergency managers during Hurricanes Irma and Maria in 2017.

(4)    Inter-Comparison of Hypoxia Models for the Northern Gulf of Mexico

The Louisiana/Texas continental shelf in the Northern Gulf of Mexico (NGOM) supports the largest hypoxic zone in U.S. coastal waters. This component of COMT arose in response to growing public concerns and included government partners from the U.S. Naval Research Laboratory and the Environmental Protection Agency (EPA). Hypoxia research in the NGOM examined causes of the temporal and spatial variability of low and harmful DO concentrations using the NCOM, FVCOM, and ROMS models [45]. The combination of low DO and high nitrate creates initially low DO conditions near the bottom, but these are not hypoxic. High phytoplankton concentrations produced in response to the input of nitrogen and other macronutrients to shelf surface waters, if not grazed, sink to the bottom, where it decomposes, consuming oxygen and creating bottom hypoxia. Other physical factors impacting hypoxia include stratification of Gulf waters, which amplifies the effects of excess nitrogen from the Mississippi-Atchafalaya River Basin [48]. The models have supplemented research cruises to sample the NGOM Dead Zone and improve nitrogen management, including agricultural management and the need for wetland and riparian buffer restoration. Some of the most recent modeling results from this study suggest that the climate changes that are underway probably exacerbate eutrophication and hypoxia [51,52].

(5)    COMT Cyberinfrastructure

The cyberinfrastructure component of COMT was designed to facilitate collaboration across the various institutions and modeling teams by enabling exploration, storing research results, and providing community access and tools. The cyberinfrastructure provides a robust data management backbone for the entire project. This backbone includes maintenance of the COMT data archive (i.e., thematic real-time environmental distributed data services—THREDDS) and web services (OPeNDAP, Sci-WMS), coordination with the modeling teams to facilitate upload of data into the data archive and address individual team challenges, review and maintenance of project metadata, development and maintenance of the COMT model viewer, implementation of a data upload tool to streamline the data ingest process, and development of a catalog tool for browsing and searching data. Details on open-source web services for COMT are described by [53].

The model viewer provides interactive access to archived COMT model data utilizing Sci-WMS, a COMT-supported, Python-based web mapping service for visualizing geospatial data. An animated example of storm wave visualization for Puerto Rico during Hurricane Georges in 1998 is offered in [39] and can be accessed at https://eos.org/science-updates/a-test-bed-for-coastal-and-ocean-modeling. While individual researchers have their own visualization and analysis tools for use with their particular numerical models, the model viewer facilitates collaboration by allowing simultaneous visualization of results from different models through inter-model comparisons. This tool extends the value of COMT results to future modeling research and development activities. The COMT model viewer can be accessed at http://comt.ioos.us/map/.

### 3.3. COMT: A Collaboratorium

Since 2010, COMT has facilitated collaboration involving more than 20 universities, industry firms, and agency representatives from NOAA, the Navy, EPA, and the U.S. Army Corps of Engineers; COMT is now one of twelve official NOAA testbeds (www.testbeds.noaa.gov). Collaboration has underpinned program success from the outset, within the academic community and, importantly, between academia, corporations, and operational users. Margaret Davidson, the former director of NOAA's Office of Coastal Management, advocated for creating a collaborative virtual community or "collaboratorium." SURA promoted such a virtual infrastructure for COMT researchers. Partners such as LSU helped SURA to leverage storage, networks, HPC, heterogeneous multi-provider services

integration, datacentric service models, and security for data processing and storage. The COMT collaboratorium IT support team collected and processed high-volume and diverse data. COMT has showcased that scientific collaboration requires dedicated effort and mutual acceptance of goals, critical assessments of diverse approaches, iterative updates, incremental improvements to understanding, promotion of new paradigms, and effective communication with a hierarchy of researchers and operational users. SURA's COMT project was able to unite government researchers, technologists, and faculty who wanted to explore innovative ways to transition research to operations.

Since September 2018, NOAA's U.S. IOOS Program Office has managed all elements of what is now called the IOOS Coastal and Ocean Modeling Testbed. Program participants of the NOAA sponsored projects provide seminal datasets that are accessible from a THREDDS server and the COMT website (see https://comt.ioos.us/). Seminal datasets mark substantial progress in a COMT project, as determined by the COMT Principal Investigator. They highlight advancement in model development or new capabilities, which are also described in the scientific literature. For example, input data sets that were used with the four-dimensional variational (4DVAR) data assimilation technique. Operational organizations remain a key beneficiary for programs such as COMT that allow researchers using differing methodologies to analyze common data sets and inter-compare model-based coastal ocean state estimates using standardized assessment metrics.

## 4. New Perspectives and Lessons Learned

SCOOP and COMT have demonstrated that collaborations involving multiple organizations and stakeholders can be effective in moving modeling innovations forward and transitioning research to operations. Numerous investigators [5,54–60] continue to highlight the importance of team science modeling experiments that are supported by real-time data and software tools that can be used across multiple scales and physical phenomena. For this reason, SURA's approach has included top-down and bottom-up communication to ensure that all participants were aware of progress, challenges, and achievements. Key lessons learned are as follows:

(1) Collaboration among team members and across projects facilitated innovation. SURA held annual principal investigator and partner meetings where researchers were able to work with operators to address issues such as the consequences of missing physics in models and the efficacy of compensatory parameterizations. In some situations, traditional diagnostics were found to be inadequate to fully assess errors in modeling processes and new approaches were developed. We expect that the role of balanced collaboration and completion for spurring innovation will carry over into future testbeds.

(2) The use of open standards and engaging with the OGC facilitated the integration of data from different regional observatories. It also helped in the discovery of observations and model output necessary for the real-time visualization of satellite images, sensors, and numerical data using common GIS applications.

(3) Collaboration among academia, government, and industry was effective, as evidenced by the use of results from SLOSH and ADCIRC (coupled with spectral wave models) to support emergency managers during Hurricanes Irma and Maria during September and October 2017. Whereas SCOOP and COMT have exemplified a transdisciplinary approach to research, further improvements are required to involve the social sciences. International researchers involved in resilience studies have described some strategies and challenges for future transdisciplinary collaborations in coastal science [61].

(4) Although progress has been made in understanding connections between physical and ecological processes, understanding of coupled natural and social systems has largely been lacking. Combining new observational strategies and modeling techniques must include factors related to economics, populations, and disease in relation to natural processes.

(5) The need continues for multi-agency investment for linked observations and modeling to predict hazards associated with severe storms, shoreline erosion, and sea level rise. Operational organizations such as NOAA and the Navy require ongoing advances in today's modeling infrastructure, including

in situ sensor networks, improved process representations, incorporation of data assimilation, better model coupling, and testing of real-time models.

(6) Networks such as U.S. IOOS provide observations that can be used to compare and improve models in a variety of coastal zones. However, demonstrations of modeling output to meet the information needs of decision makers should be planned prior to the occurrence of a natural disaster. Catastrophic flooding events that have followed extreme rainfall, such as in East Baton Rouge during August 2016, demonstrate the need for improvements in the coupling of hydrology and ocean models and expansion in the hierarchy of available models and observations to help quantify sources of uncertainty in simulations.

(7) Continued research is needed to determine how initial state, coupling of system components, and changes in external forcing contribute to predictability. It is apparent through COMT that predictions needed to support many decision-makers require increased resolution using HPC and "Big Data" resources. COMT has illustrated that seamless prediction from minutes to decades is model- and resolution-dependent and that the process of understanding model errors and determining optimal resolution and ensemble size to predict useful measures, such as means or extremes, can only be advanced through collaboration between researchers and operators.

(8) Some of the operational needs of federal, state, and local organizations responsible for emergency response, coastal protection, resource management, research, and national defense are described in [62]. The Navy needs to model the coastal ocean to maintain control of vital areas of the ocean and numerical modeling of environmental factors that impact naval operations. NOAA requires improved understanding of the connections among storms, hazards, society, and ecosystems. The U.S. Army Corps of Engineers partnered with several COMT projects and benefits through improved data and models to operate hundreds of coastal ports and navigation channels throughout the nation. States bordering the Chesapeake Bay and the Gulf of Mexico must cope with and mitigate the effects of hypoxia contributing to dead zones. SCOOP and COMT have improved modeling capabilities and expanded environmental intelligence for use by decision-makers and thereby have helped to create more resilient coastal communities [63].

(9) An important component of the COMT project was the effort in sharing HPC resources, knowledge transfer, and especially the implementation of advances made by the COMT cyberinfrastructure team. These accomplishments provide the basis for the next generation of COMT projects and soon to be implemented operational programs within the NOAA-IOOS structure.

## 5. Conclusions and Prognosis

What we have described in this paper has been the evolution of a methodology that is underpinned not by new technology but predominantly by collaboration. The National Research Council reported, "Without a strong, effective collaboration among the government, academic, and private sectors, the general public would not have been the beneficiary of the great advances in weather and climate science and technology over the last 50 years" [64]. Development and implementation of SCOOP and COMT has involved university researchers, government scientists, private industry, community-based organizations, and policy-makers working together to (a) define the problems to be addressed and the questions that need to be answered; (b) evolve and improve modeling methodologies within a collaborative environment; (c) interpret research results in terms of their significance for community and policy change, and (d) disseminate the research findings and advocate for change. Truly collaborative teams respect the knowledge and contributions that each partner brings to the project to understand the complex problems facing coastal communities so that science-based responses to those problems can be implemented. There is a variety of multi-disciplinary approaches currently used on large projects. Three elements are central to them all: (1) diversity, (2) collaboration, and (3) a community-based focus. Where numerical models are concerned, community-based implies open source models. Projects such as COMT connect researchers from multiple interdisciplinary fields to solve complex problems. Innovations can often result from the mathematician, biologist, oceanographer, and economist working

together in an integrated way. Co-responsibility must be created between researchers and stakeholders. Whereas collaboration cannot completely dissolve competitive issues and constraints of university hierarchies, a healthy balance stimulates the development of alternatives and new analytical techniques and also challenges the social, economic, and political status quo. In the case of COMT, the research has met needs for community modelers and operational sponsors. These central elements were the foundation for transdisciplinary research accomplished through SCOOP and COMT.

Collaboration is now needed on a larger scale to address many compelling grand challenges emerging from climate change. Extensive collaboration will be required in order to develop coupled models of hydrologic and ocean flooding in order to understand the probabilities and effects of compound inundation events. Understanding and predicting the existing and future connections among physical, ecologic, and socioeconomic factors pose even greater challenges. This is particularly the case for communities in low elevation coastal zones that are vulnerable now and will be much more vulnerable in future decades. Prominent examples prevail along the low-lying coast of the Gulf of Mexico, where storm surges are significantly amplified by the wide low-gradient continental shelf (Figure 3) and vanishing wetlands provide the only protection for low-income communities. With specific reference to the Gulf Research Program, the National Academies [65] recently recommended the creation of " . . . a resilience learning collaborative for stakeholders to exchange information, approaches, challenges, and successes in their respective and collective work to advance community resilience in the Gulf region. The collaborative participants should include government (local, state, federal levels), industry, academia, and other organizations engaged in addressing community resilience". As pointed out in a recent international white paper on modeling and observations to support coastal resilience [62], "Linked observing and modeling programs that involve stakeholder input and integrate engineering, environmental, and community vulnerability are needed to evaluate conditions prior to and following severe storm events, to update baselines, and to plan for future changes over the long term."

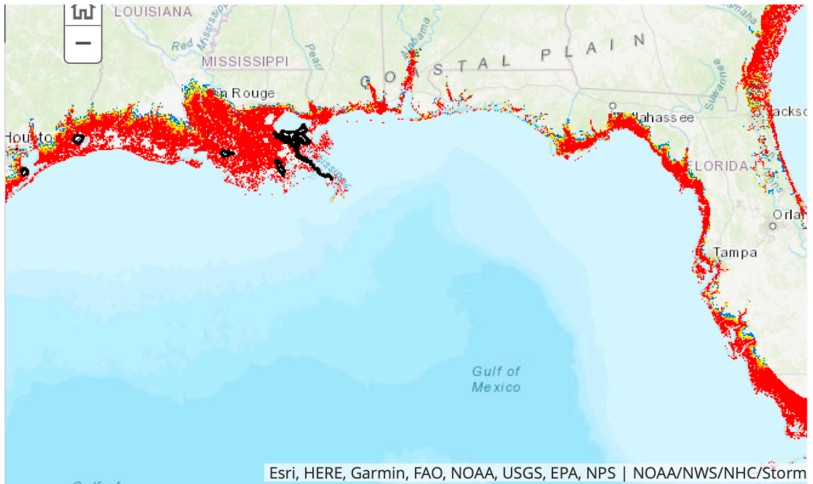

**Figure 3.** National Hurricane Center SLOSH predictions of the "maximum of maximum envelopes of water" (MOMS) caused by storm surges associated with landfalling Category 5 hurricanes on the central and eastern regions of the Gulf Coast (see https://www.nhc.noaa.gov/). The red areas could expect inundation to exceed 9 feet (~3 m) above existing ground levels. Levees currently protect areas shown in black.

**Author Contributions:** C.R.N. and L.D.W. synthesized the literature and wrote this review. Both authors also helped coordinate and manage the SCOOP and COMT programs. All authors have read and agree to the published version of the manuscript.

**Funding:** The collaborative research reviewed here was supported by the Southeastern Universities Research Association (SURA), the Office of Naval Research (Grant Number N00014-04-1-0721), and both the NOAA

Coastal Services Center and U.S. IOOS Program Office (Award Numbers NA04NOS4730254, NA10NOS0120063, NA11NOS0120141, and NA13NOS0120139).

**Acknowledgments:** This work benefited from support and guidance from Jerry Draayer (SURA), Mel Briscoe (ONR), Margaret Davidson (NOAA), and Zdenka Willis (US IOOS). Project leaders, principal investigators, and program managers at different stages of the program played crucial roles and deserve special acknowledgement. In alphabetical order, these essential contributors were as follows: Luis Bermudez (Open Geospatial Consortium), Joanne Bintz (SURA, Deceased), Philip Bogden (University of Maryland Baltimore County), Gary R. Crane (SURA), Richard A. Luettich, Jr. (University of North Carolina Chapel Hill), Donald T. Resio (University of North Florida), Elizabeth Smith (Old Dominion University), and Gary A. Zarillo (Florida Institute of Technology). We also thank Frank Aikman from NOAA, Becky Baltes, a NOAA affiliate, Eric Baylor from NOAA, Rich Signell from USGS, and Aijun Zhang from NOAA. Key project leaders who collaborated with fellow researchers and operational partners included Katja Fennel from Dalhousie University, Marjorie A.M. Friedrichs from Virginia Institute of Marine Science, Sara Graves from University of Alabama in Huntsville, Eoin Howlett from RPS Group, Alexander Kurapov from Oregon State University, and Andre van der Westhuysen, a NOAA affiliate. We thank our colleagues from SURA and affiliated organizations who have provided insight and expertise that greatly assisted in the development of SCOOP and COMT. Thoughtful insights were contributed by SURA Fellows Carolyn Thoroughgood from the University of Delaware and Robert Weisberg from the University of South Florida. Information technology tasks by Hortensia T. Valdes from LSU and HPC coordination by Linda Akli from SURA benefited all parties, as well as legal and contract management tasks and meeting facilitation accomplished by SURA staff members Peter Bjonerud, John Holly, Sara Madden, Russell Moy, A'fenia Pirtle-Hubbard, and William Jones.

**Conflicts of Interest:** The authors declare no conflict of interest.

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
