# Peer review of "The Evolution and Outcomes of a Collaborative Testbed for Predicting Coastal Threats"

_jmse, doi:10.3390/jmse8080612_

Round 1

Reviewer 1 Report

Beginning in 2003 the Southeastern Universities Research Association (SURA) enabled an open15 access network of distributed sensors and linked computer models through the SURA Coastal Ocean Observing and Predicting (SCOOP) program. The goal was to support collaborations among universities, government, and industry to advance integrated observation and modeling systems. SCOOP improved the path to operational real-time data-guided predictions and forecasts of coastal ocean processes. This was critical to the maritime infrastructure of the U.S. and to the well-being of coastal communities. SCOOP integrated and expanded
observations from the Gulf of Mexico, the South Atlantic Bight, the Middle Atlantic Bight, and the Chesapeake Bay. From these successes, a Coastal and Ocean Modeling Testbed (COMT) evolved with NOAA funding via the Integrated Ocean Observing System (IOOS) to facilitate the transition of key models from research to
 operations. Since 2010, COMT has been a conduit between the research community and the federal government for sharing and improving models and software tools. SCOOP and COMT have been based on strong partnerships among universities and U.S. agencies that have missions in ocean and coastal environmental prediction. During SURA’s COMT project, significant progress was made in evaluating the performance of models that are progressively becoming operational.

Author Response

Thank you for your review. I do not see any specific comments other than a copy of the abstract. However, I will note that the following revisions have been made to the paper: 1. We have added 15 new references specifically in respect to Hurricane Katrina, specific models and their inter comparisons and plans for the future. 2. We deleted 3 key words that did not appear elsewhere in the text and replaced them with other more relevant key words that did appear in the text. 3. We edited and in some cases completely rewrote several paragraphs describing COMT results from specific projects to make them clearer and avoid repetition. 4. In the conclusions and prognosis section, we added a figure showing flooding flooding potential on the Gulf of Mexico coast and added text advocating for future programs to address the socioeconomic impacts for such flooding. If we have missed some intended comments or suggestions, please send them to us after examining the revised paper and we will address them. 

Reviewer 2 Report

Thank you for giving me such an opportunity to review this paper. Authors stated the evolution and outcomes of a collaborative work for predicting the coastal threats in US and listed several new perspectives and lessons learned. Such a kind of joint work will be needed in the future in other countries. I do not have too many comments. The only comment that I want to share is that you can add some references for the models such as in line 96 and 97. Moreover, paper citations are also needed in Line 136 for the model grid testing for Hurricane Katrina track.

Author Response

Thank your your review. We agree with your suggestions and have added appropriate references to the paper, including with regard to H. Katrina.  I  note that the following revisions have been made to the paper: 1. We have added 15 new references specifically in respect to Hurricane Katrina, specific models and their inter comparisons and plans for the future. 2. We deleted 3 key words that did not appear elsewhere in the text and replaced them with other more relevant key words that did appear in the text. 3. We edited and in some cases completely rewrote several paragraphs describing COMT results from specific projects to make them clearer and avoid repetition. 4. In the conclusions and prognosis section, we added a figure showing flooding potential on the Gulf of Mexico coast and added text advocating for future programs to address the socioeconomic impacts for such flooding. 

Reviewer 3 Report

Informative and useful description of general aspects, goals and objectives of SCOOP and COMT projects. The article is well within the scope of the journal. The article does not set out a new scientific or engineering result, but it will undoubtedly arouse the interest of the reader.

Remaks

  1. Lines 28-30. Some Keywords are missing in the text: Grid Computing; High Performance Computing; coastal inundation.
  2. L. 185-186. Table 2. To correct panel Description. Used "unequal", different characteristics of the models. FVCOM - free-surface, 3-D primitive equation… almost all circulation models are «free-surface»; HYCOM – primitive equation (PE) general circulation model – almost all models are PE, indicate feature of HYCOM – vertical coordinate, etc; SELFE - model for cross-scale ocean modeling – what does “cross-scale” mean?; not only WW3 - third-generation wave model SWAM also, or only WW model is understood here?, etc.
  3. L. 210-224. It would be nice to add some digital characteristics of the described results.
  4. L. 394-453. In my opinion, the article will improve if the authors add a picture of a bright oceanographic result - circulation or something else, plus formulate a new achievement.

Author Response

Thank you for your review. We have attempted to address all of your helpful suggestions with exception of point 3 which we felt would add detail beyond the general scope which we aimed at a broad audience. However, we added several references which guide the readers to those details.We also deleted unnecessary terms such as "free surface" from Table 2 and replaced "different" with "various". I will note that the following additional revisions have been made to the paper: 1. We have added 15 new references specifically in respect to Hurricane Katrina, specific models and their inter comparisons and plans for the future. 2. We deleted 3 key words that did not appear elsewhere in the text and replaced them with other more relevant key words that did appear in the text. 3. We edited and in some cases completely rewrote several paragraphs describing COMT results from specific projects to make them clearer and avoid repetition. 4. In the conclusions and prognosis section, we added a figure showing flooding  potential on the Gulf of Mexico coast and added text advocating for future programs to address the socioeconomic impacts for such flooding. If we have missed some intended comments or suggestions, please send them to us after examining the revised paper and we will address them.